# Dietary Vitamin D Mitigates Coronavirus-Induced Lung Inflammation and Damage in Mice

**DOI:** 10.3390/v15122434

**Published:** 2023-12-15

**Authors:** Gabriel Campolina-Silva, Ana Cláudia dos Santos Pereira Andrade, Manoela Couto, Paloma G. Bittencourt-Silva, Celso M. Queiroz-Junior, Larisse de Souza B. Lacerda, Ian de Meira Chaves, Leonardo C. de Oliveira, Fernanda Martins Marim, Cleida A. Oliveira, Glauber S. F. da Silva, Mauro Martins Teixeira, Vivian Vasconcelos Costa

**Affiliations:** 1Department of Morphology, Institute of Biological Sciences, Universidade Federal de Minas Gerais, Belo Horizonte 30270-901, MG, Brazil; anaclaudiaandrade29@gmail.com (A.C.d.S.P.A.); larissesbl.adv@gmail.com (L.d.S.B.L.); iandemeirachaves@hotmail.com (I.d.M.C.); cleida@icb.ufmg.br (C.A.O.); 2Department of Biochemistry and Immunology, Institute of Biological Sciences, Universidade Federal de Minas Gerais, Belo Horizonte 30270-901, MG, Brazil; ldeolive2@gmail.com (L.C.d.O.); mmtex.ufmg@gmail.com (M.M.T.); 3CHU de Québec Research Center (CHUL), Université Laval, Quebec, QC G1V 4G2, Canada; 4Department of Physiology and Biophysics, Institute of Biological Sciences, Universidade Federal de Minas Gerais, Belo Horizonte 30270-901, MG, Brazilglauber@ymail.com (G.S.F.d.S.); 5Department of Genetics, Ecology and Evolution, Institute of Biological Sciences, Universidade Federal de Minas Gerais, Belo Horizonte 30270-901, MG, Brazil

**Keywords:** Vitamin D, betacoronavirus, SARS-CoV-2, murine coronavirus (MHV), respiratory infection

## Abstract

The COVID-19 pandemic caused by the SARS-CoV-2 (β-CoV) betacoronavirus has posed a significant threat to global health. Despite the availability of vaccines, the virus continues to spread, and there is a need for alternative strategies to alleviate its impact. Vitamin D, a secosteroid hormone best known for its role in bone health, exhibits immunomodulatory effects in certain viral infections. Here, we have shown that bioactive vitamin D (calcitriol) limits in vitro replication of SARS-CoV-2 and murine coronaviruses MHV-3 and MHV-A59. Comparative studies involving wild-type mice intranasally infected with MHV-3, a model for studying β-CoV respiratory infections, confirmed the protective effect of vitamin D in vivo. Accordingly, mice fed a standard diet rapidly succumbed to MHV-3 infection, whereas those on a vitamin D-rich diet (10,000 IU of Vitamin D_3_/kg) displayed increased resistance to acute respiratory damage and systemic complications. Consistent with these findings, the vitamin D-supplemented group exhibited lower viral titers in their lungs and reduced levels of TNF, IL-6, IL-1β, and IFN-γ, alongside an enhanced type I interferon response. Altogether, our findings suggest vitamin D supplementation ameliorates β-CoV-triggered respiratory illness and systemic complications in mice, likely via modulation of the host’s immune response to the virus.

## 1. Introduction

Vitamin D, obtained via UVB radiation exposure, dietary intake, or supplementation, assumes a pivotal role as a secosteroid hormone in the human body. Once converted into its active form, 1,25-dihydroxyvitamin D (also known as calcitriol), it binds to the vitamin D receptor (VDR) to modulate gene expression in a cell- and species-specific manner [1,2]. Besides its classic role in calcium and bone metabolism, vitamin D influences cellular responses during infections by modulating the expression of antimicrobial peptides [3,4,5] and enhancing type I interferon responses [6,7]. Vitamin D metabolites can also influence innate and adaptive immune responses by modulating cell proliferation and differentiation of leukocytes, cytokine responses, and reducing toll-like receptor (TLR) activation. These immunoregulatory actions are well appreciated in the attenuation of injuries caused by uncontrolled immune response [8]. In line with these findings, epidemiological studies have pointed out an existing link between low serum levels of 25-hydroxyvitamin D [25(OH)D], the major circulating metabolite of vitamin D, and increased risk of respiratory infections, including influenza [9], pneumonia [10], and tuberculosis [11]. Furthermore, vitamin D supplementation has been shown to reduce the overall risk of acute respiratory infections in humans [12].

The COVID-19 pandemic has prompted extensive exploration into the immunoregulatory properties of vitamin D, yielding a complex and often contradictory landscape of findings. Numerous observational studies have proposed a connection between vitamin D deficiency or insufficiency and adverse COVID-19 outcomes, emphasizing a potential role in disease susceptibility or severity [13]. However, a counter-narrative has emerged from other studies, which fail to establish a significant association between predicted 25(OH)D levels and COVID-19 incidence or severity [14]. The dichotomy persists in randomized controlled trials, where a subset of studies suggests that vitamin D supplementation reduces SARS-CoV-2 infection [15], as well as hospitalization and mortality rates [7,16], while others report no discernible impact on these clinical endpoints or disease incidence [17,18]. Challenges in reconciling these divergent findings may stem from variations in study methodologies, population demographics, dosages, and baseline vitamin D statuses, emphasizing the need for meticulous consideration of these factors in the interpretation of results and the formulation of public health recommendations. For a comprehensive examination of the subject matter, recent systematic reviews and meta-analyses should be consulted [19,20].

While important randomized clinical trials are underway to provide more clarity on the topic (e.g., NCT04536298, NCT04868903, NCT04482673, NCT04385940, and NCT03188796), there remains a lack of preclinical evidence supporting the prophylactic or therapeutic roles of vitamin D against SARS-CoV-2 infection. Initially, it was shown that a vitamin D-induced antimicrobial peptide, cathelicidin LL-37, could inhibit viral adsorption and host cell entry of SARS-CoV-2 [21]. Later on, Mok et al. (2023) identified calcitriol as a promising inhibitor of SARS-CoV-2 replication in vitro, in part by means of upregulating LL-37 expression. However, both vitamin D and calcitriol failed to protect K18-hACE2 mice and hamsters from SARS-CoV-2 infection [22,23]. Disparities between in vitro and in vivo results may be either due to the absence of the vitamin D response element (VDRE) in the promoter region of genes encoding antimicrobial peptides in rodents or the likelihood of low-dosage delivery of vitamin D metabolites [2,4]. Thus, understanding the antiviral properties of vitamin D against respiratory infections caused by SARS-CoV-2 or other β-coronaviruses (β-CoVs) remains a focus of future preclinical research.

In this study, we strengthened the body of evidence supporting the antiviral properties of vitamin D. Our research demonstrates that calcitriol effectively reduces the in vitro replication of SARS-CoV-2 as well as murine β-CoVs (MHV-3 and MHV-A59). Furthermore, we showed that dietary vitamin D supplementation significantly mitigates lung inflammation, tissue damage, and lethality in mice intranasally infected with MHV-3, a murine model that mirrors the acute respiratory syndrome and systemic complications usually seen in severe cases of COVID-19 [24,25,26].

## 2. Materials and Methods

### 2.1. Virus Propagation and Plaque Assay

MHV-3 (GenBank accession no. MW620427.1; Ref. [27]) and MHV-A59 (ATCC VR-764^TM^) was amplified in L929 cells under a controlled atmosphere (37 °C, 5% CO_2_). Briefly, L929 cells grown in T-75 flasks were inoculated with serum-free Dulbecco’s Modified Eagle Medium (DEMEM, Cultilab, Campinas, Brazil) containing each virus at a multiplicity of infection (MOI) of 0.1. After 24 h, the virus-containing supernatants were collected and concentrated using Amicon^®^ Ultra-4 Centrifugal Filter Units (Millipore, Burlington, MA, USA). The MHV titer was determined via plaque assay of L929 cells seeded at 1 × 10^5^ density onto 24-well plates [24].

Parental SARS-CoV-2 Wuhan-Hu-1 (GenBank accession no. NC_045512.2; Ref. [28]) was propagated in Vero CCL-81^TM^ (ATCC, Manassas, VA, USA). For virus stock production, Vero CCL-81 cells were kept in serum-free DEMEM and inoculated with SARS-CoV-2 (MOI 0.1) for 24 h before collecting the supernatant. All experiments with infectious viruses were carried out in a biosafety level 3 (BSL3) multiuser facility, in accordance with WHO guidelines. SARS-CoV-2 titer was determined as p.f.u/mL. To this end, monolayers of Vero CCL-81 (1 × 10^5^ cell/well) in 24-well plates were infected with serial dilutions of supernatants containing SARS-CoV-2 for 1 h at 37 °C. Fresh semisolid medium containing 1.2% carboxymethylcellulose (CMC) was added, and the culture was maintained for 96 h at 37 °C. Cells were fixed with 10% formaldehyde solution for 1 h at room temperature and then stained with crystal violet (0.5%) for an additional 0.5 h.

### 2.2. Primary Cultures

Primary cultures of bone marrow-derived macrophages (BMDMs) were established from 7–10-week-old C57BL/6J mice, as previously described [29]. On the 6th culture day, BMDMs were resuspended in L929-conditioned medium (5% L929 supernatant, 10% fetal bovine serum [Gibco, Carlsbad, CA, USA] and 2 mM L-glutamine [Gibco, USA] in RPMI 1640 [CultLab, São Paulo, Brazil]) and seeded at 2.5 × 10^5^ cell density onto 24-well plates. On the next day, BMDMs were washed with PBS and inoculated with 150 µL serum-free RPMI 1640 loaded with either MHV-3 or MHV-A59 at 0.1 MOI. After 1 h of viral adsorption, cells were washed in PBS and cultured for 24 h with 0.5 mL of fresh RPMI 1640 containing calcitriol (Cayman Chemical Co., Ann Arbor, MI, USA) at different concentrations (0.1, 1.0, 2.5, or 5.0 µM) or 0.01% ethanol at 100% (vehicle, corresponding to the 5.0 µM dose of calcitriol). Virus yields were quantified in the supernatant via plaque assay. Cell damage was assessed indirectly by quantifying the abundance/activity of lactate dehydrogenase (LDH) in the supernatant via spectrophotometric measurements (catalog no. K014, Bioclin, Belo Horizonte, Brazil).

### 2.3. SARS-CoV-2 Analysis in Calu-3 Cells

Calu-3 cells were infected with SARS-CoV-2 at an MOI of 0.1. Cells were infected at densities of 1 × 10^5^ cells/well in 24-well plates for 1 h at 37 °C. After 1 h, the cell monolayers were washed, and 1 mL DMEM supplemented with calcitriol at different concentrations (0.1, 1.0, 2.5, 5.0, 7.5, and 10 µM) was added. Calu-3 cells infected and untreated or treated with vehicle only (0.02% ethanol 100%) were kept as controls. After 48 h, the supernatants were collected, and virus yields and LDH activity were determined via plaque assay and spectrophotometrically, respectively.

### 2.4. Animals and In Vivo MHV-3 Infection

Animal experimental procedures were reviewed and approved by the Ethical Committee for Animal Experimentation of the Universidade Federal de Minas Gerais—UFMG. C57BL/6J mice aged 4 weeks (n = 30 females and 30 males) were purchased from the UFMG Specific-pathogen-free (SPF) Animal House and kept under SPF conditions and controlled temperature (24 °C ± 2 °C) and light cycle (12 h light/12 h dark). At 6 weeks of age, half the mice (n = 15 females and 15 males) were given ad libitum access to standard rodent chow (1000 UI vitamin D_3_/kg, AIN-93M formulation, PragSoluções Biociências, Jaú, Brazil), while the others had access to a vitamin D_3_-supplemented diet (10,000 UI/kg, AIN-93M formulation, PragSoluções Biociências, Brazil). At 8 weeks of age, mice were inoculated intranasally with 30 µL sterile saline solution loaded or not (mock controls) with 10^3^ plaque-format units (p.f.u) of MHV-3 and monitored for up to 14 days post inoculation (dpi), as described previously [24] (Andrade et al. 2021). Diet intervention stopped after the first week post infection to avoid hypervitaminosis D in the group fed a vitamin D-rich diet. Disease severity was assessed daily and graded blinded in each of the following categories: hunched posture (0–2), ruffled fur (0–2), rapid shallow breathing (0–2), squinted eyes/conjunctivitis (0–3), lethargy (0–3), and cachexia (0–3). Daily scores (sum of all categories) were calculated for each mouse and averaged across the groups.

### 2.5. Sample Collection and Processing

Mice were euthanized at 3 or 5 dpi via deep anesthesia (ketamine [80 mg/kg]-xylazine [10 mg/kg], i.p.) followed by cervical dislocation. Blood samples were collected from the abdominal vena cava, and the white blood cell count was estimated using the Celltac MEK-6500K hemocytometer (Nihon Kohden, Tokyo, Japan). The right lobes of the harvested lungs were snap-frozen in liquid nitrogen, while the left lobes were fixed in 10% neutral buffered formalin for 24 h and then embedded in paraffin (FFPE samples). In addition, the liver and spleen were collected and processed accordingly.

### 2.6. Histopathology

Tissue damages caused by MHV-3 infection were assessed at 3 or 5 dpi in FFPE lung and liver sections (5 µm thickness) stained with hematoxylin and eosin. The analyses followed the grading system established by Andrade et al. (2021) [24] and were performed by an experienced pathologist (C.M.Q.-J.), who was blinded to the experimental groups.

### 2.7. Immunofluorescence and Confocal Microscopy

Immunofluorescence assays were carried out to detect the pan-leukocyte marker CD45 and cleaved caspase 3 (apoptosis marker) in 5-µm FFPE lung sections of mice at 3 dpi. Following heat-induced antigen unmasking (10 min boiling in 10 mM sodium citrate, pH 6.0), sections were permeabilized in PBS/0.5% Triton X-100 (15 min) and then blocked for 1 h with 5% goat serum and mouse BD Fc Block (5 mg/mL). Sections were labeled overnight at 4 °C with rabbit anti-cleaved caspase 3 (1:200, catalog no. 9664S, Cell Signaling Technology, Danvers, MA, USA) and with APC-conjugated rat anti-mouse CD45 antibody (1:100, catalog no. 559864; BD Pharmingen, San Diego, CA). For apoptosis detection, sections were additionally incubated with AlexaFluor^TM^ 488-conjugated goat anti-rabbit antibody (1:400, catalog no. A28175) for 1 h. Cell nuclei were stained for 5 min with DAPI (1 µg/mL). The sections were analyzed in a Nikon A1-HD laser scanning confocal microscope.

### 2.8. Cytokines, Calcium and 25(OH)D Measurements

The mouse DuoSet ELISA system (catalog no. DY401, DY406, DY410, DY485, R&D Systems, Minneapolis, MN, USA) was used to determine the concentrations of TNF, IFN-γ, IL-6, and IL-1b in mouse lung homogenates at 3 dpi. Samples were processed as described previously [24], and the assays were performed following the manufacturer’s recommendations.

Serum 25(OH)D levels (including 25-diidroxivitamin D_2_ and D_3_) were measured using the DRG 25-OH-Vitamin D ELISA kit (catalog no. EIA-5396), as previously described [30]. Serum calcium was determined via a colorimetric assay based on the Arsenazo III reaction system (catalog no. K051, Bioclin, Belo Horizonte, Brazil).

### 2.9. Quantitative Reverse Transcription Polymerase Chain Reaction (RT-qPCR)

Total RNA was isolated from 20 mg of lung tissue using the Aurum^TM^ RNA Kit (Catalog No. 7326820, Bio-Rad, Hercules, CA, USA) and then reverse-transcribed into cDNA with the iScript^TM^ cDNA Kit (Catalog No. 1708890, Bio-Rad). RT-qPCR was performed on a 7500 Fast Real-Time PCR System (Thermo ABI, Waltham, MA, USA) using a 10 μL PowerTrack™ SYBR™ Green Master Mix (Catalog No. A46109, Applied Biosystems, Waltham, MA, USA), with 2 ng of cDNA and 0.5 μM of both forward and reverse primers (Table 1). mRNA levels for *Ceacam 1*, *Defb1*, *Ifnb1*, *Isg15*, and *Isg20* were calculated via the 2^−ΔΔCt^ method, using the geometric mean of *Gapdh* and *Actb* values for normalization [31]. Results were expressed as fold change.

### 2.10. Statistical Analysis

All data were checked for normal distribution with the Shapiro–Wilk test and Q-Q plots. Unless indicated, Student’s *t*-test or ordinary one-way ANOVA plus Fisher’s LSD post hoc test were used to compare means between two or more groups, respectively. Non-parametric comparisons were carried out using the Mann-Whitney U test or Kruskal–Wallis plus Dunn’s post hoc test, when two or more groups were considered, respectively. Survival rates among groups were evaluated via Kaplan–Meier survival analysis. All analyses were performed in the GraphPad Prism 9.0 software, using *p* < 0.05 as the significance threshold. In addition to individual *p* values calculated using post hoc tests, we employed the Benjamini, Krieger, and Yekutieli method to provide false discovery rate (FDR)-adjusted *p* values. Raw data and statistics were made available in Appendix A. 

## 3. Results

### 3.1. Vitamin D in Its Biologically Active form Reduces Replication of Coronaviruses In Vitro

To first assess the potential antiviral property of vitamin D against β-CoVs, we exposed bone marrow-derived macrophages (BMDMs) to two murine β-CoV strains, MHV-3 and MHV-A59, prior to treating the cells for 24 h with increasing concentrations of calcitriol (Figure 1a). Calcitriol is the active form of vitamin D that acts via the vitamin D receptor (VDR), a transcript factor known to be expressed in BMDMs [32]. As shown in Figure 1b,d, post-infection treatment with 5 μM calcitriol resulted in a significant reduction in MHV-3 titer by 2.1 log_10_ p.f.u/mL (*p* < 0.0001), as well as in MHV-A59 titers (1.3 log_10_ p.f.u/mL reduction; *p* = 0.0002). Accordingly, calcitriol at 5 μM also reduced virus-induced cell damage in BMDMs (Figure 1c,e).

Next, we tested the effect of calcitriol on the human airway cell line Calu-3 upon SARS-CoV-2 infection, a suitable in vitro model for antiviral testing against SARS-CoV-2 [33] and which also expresses VDR [34]. Accordingly, treatment with calcitriol led to a gradual, significant decrease of SARS-CoV-2 viral load when cells were treated with calcitriol at 5 μM (1.3 log_10_ p.f.u/mL reduction; *p* = 0.0108) or higher (e.g., 10 μM; 2.3 log_10_ p.f.u/mL reduction; *p* = 0.0002) (Figure 1f). Calcitriol treatment did not alter SARS-CoV-2-induced cell damage, as LDH release/activity was similar among groups (Figure 1g). This is in line with a recent report showing that calcitriol treatment has a limited effect on preventing cell damage induced by SARS-CoV-2 in vitro infection [23]. Of note, calcitriol was well tolerated at higher concentrations (7.5 μM and 10 μM) by Calu-3 but not by BMDMs (Appendix A). These results suggest an anti-β-CoV activity of vitamin D that warrants further investigation in vivo.

### 3.2. Vitamin D Decreases Susceptibility of Wild-Type Mice to Severe MHV-3 Infection

Next, we tested the potential antiviral effects of vitamin D in vivo. To this end, C57BL/6J wild-type mice were intranasally infected with MHV-3, a model for studying the pathogenesis of β-CoV-induced respiratory disease [24]. Mice were fed a vitamin D_3_-rich diet (10,000 UI/kg diet; Vit. D group) over two weeks prior infection, while the control group received access to a standard diet (1000 UI vitamin D_3_/kg diet; STD group) (Figure 2a). Both diets were given ad libitum. Vitamin D_3_ was administered orally in its unprocessed form (cholecalciferol) to better mimic the common use in humans and reduce the chances of animals developing hypercalcemia, a well-documented side-effect of calcitriol [35].

The safety and efficacy of vitamin D supplementation were assessed in a pilot study (n = 5 mice per diet group). After two weeks of experimentation, we collected whole blood samples and found that levels of 25(OH)D, the main circulating metabolite of vitamin D, doubled in the serum of animals in the Vit. D group (48.84 ± 4.80 ng/mL) in comparison to the STD group (24.75 ± 3.39 ng/mL) (Table 2). Although serum calcium was found in slightly increased concentrations in the Vit. D group (6.870 ± 0.57 vs. 5.724 ± 0.35 mg/dL in the STD group; *p* = 0.1230; Table 2), it was still below the upper limit range defined for C57BL/6J mice (10 mg/dL or 2.50 nmol/L; Ref. [36]). Overall, these data suggest that the vitamin D supplementation protocol used in this study has proved to be safe for C57BL/6J mice. The significant increase in serum 25(OHD) levels in the Vit. D group was not associated with any differences in food intake or body weight gain over the experimental period (Table 2).

Following infection with 10^3^ p.f.u of MHV-3 (Figure 2a), mice fed a standard diet exhibited the typical clinical signs of disease manifestation previously described [24]. These included body weight loss, hunched posture, ruffled fur, lethargy, and rapid shallow breathing, which were first observed at 3 days post infection (dpi) and progressively worsened, leading to the mortality of all infected mice (n = 10) by the 6th dpi (Figure 2b–d). Conversely, mice in the Vit. D-supplemented group presented with less severe symptoms and showed partial protection against sudden death at the 6th dpi (70% mortality [n = 7/10], median survival = 8 dpi; *p* < 0.0001) (Figure 2c,d). Notably, 30% (n = 3/10) of mice fed a vitamin D-rich diet achieved complete recovery from the MHV-3 infection and survived until the end of the 14-day follow-up period. Furthermore, a less intense febrile response was observed in the Vit. D group upon MHV-3 infection (Figure 2e), besides exhibiting significantly lower viremia (2.8 ± 0.72 log_10_ MHV-3 p.f.u/mL plasma) than the STD group (5.0 ± 0.22 log_10_ MHV-3 p.f.u/mL plasma) at 5 dpi (Figure 2f).

### 3.3. Vitamin D Mitigates Inflammation and Acute Lung Damage Induced by MHV-3

Akin to SARS-CoV-2 [37] and other murine β-CoVs (MHV-A59 [38] and MHV-1 [39]), MHV-3 infection is known to cause leukopenia and induce a massive infiltration of immune cells in the lungs. This inflammatory response can be harmful and culminate in severe lung injuries and acute respiratory distress [24]. Therefore, we investigated the effects of vitamin D on the immune response and lung injury induced by MHV-3 infection in mice at 3 dpi. This time point corresponds to the peak of lung infection in the MHV-3 model [24].

Vitamin D markedly reduced MHV-3 titers in the lungs. Infectious virus particles could be recovered in the lung homogenates of all mice (n = 5) on the standard diet. In the Vit. D group, 3 out of 5 mice (60%) had detectable infectious virus in their lungs; however, it was at a significantly lower titer (decrease of 1.3 log_10_ p.f.u/g; *p* = 0.0079) (Figure 3a). Vitamin D also alleviates the leukopenia response linked to leukocytes trafficking to the lungs (Figure 3b), which is a hallmark of many respiratory infections [37]. A reduction in lymphocyte counts in the blood was the primary factor driving leukopenia, with the lymphocyte densities being reduced 3.8-fold in infected mice fed a standard diet and 1.75-fold in the Vit. D group (Figure 3c). Furthermore, MHV-3-infected mice in the STD group showed the highest neutrophil-to-lymphocyte ratio (NLR; infected STD vs. Vit. D. mice: 2.43 ± 0.17 vs. 1.34 ± 0.12; *p* = 0.0476) (Appendix A), a recognized indicator of poor prognosis in MHV [40,41] and SARS-CoV-2 infections [42].

This observation was consistent with the decreased infiltration of leukocytes in the peribronchial and alveolar regions in the Vit. D group, as determined via CD45 immunostainings and histological examination (Figure 3d–f). Furthermore, vitamin D mitigated the lung injury caused by MHV-3. While mice fed a standard diet had significant damage in their lungs, such as alveolar edema and necrosis, those in the Vit. D group showed lung features and histopathological scores comparable to mock controls (Figure 3f,g). An increase in caspase 3^+^apoptotic cells was observed in lung sections in the Vit. D group at 3 dpi, albeit at a much lower rate than the STD group (Figure 3h).

Next, we checked for changes in the cytokine profile in lung homogenates. We have previously shown that important pro-inflammatory cytokines, such as TNF, IL-6, IL-1β, and IFN-γ, were present at excessive concentration in the mouse lungs during MHV-3 infection and account for disease severity [24]. In line, inhibition of cytokines signaling has been shown to protect mice from MHV-3 infection [24,43], as well as mitigate damage in human cells and organs caused by SARS-CoV-2 [44,45]. As shown in Figure 4, there was an increase in the intrapulmonary concentrations of TNF, IL-6, IL-1β, and IFN-γ at 3 dpi, in the STD group. On the other hand, MHV-3-induced increases in pro-inflammatory cytokine concentrations were prevented by vitamin D supplementation, except for TNF, which occurred at increased levels compared to mock controls (Figure 4). Altogether, these results suggest that vitamin D prevents mice from developing an uncontrolled immune response and lung injuries following MHV-3 infection.

### 3.4. Vitamin D Supplementation Potentiates Type I Interferon Antiviral Responses against MHV-3

Type I interferons (IFN-I) are crucial in driving antiviral responses to host cells via the induction of interferon-stimulated genes (ISGs). It has been proposed that vitamin D supplementation significantly enhances IFN-I responses in COVID-19 patients while decreasing the risk of death [7]. Therefore, herein we examined the transcriptional expression of IFN-β (*Ifnb1*) and ISGs (*Isg15* and *Isg20*) in mouse lungs at 3 dpi, along with the levels of *Ceacam1*, which encodes for the MHV entry receptor [46,47]. Our results demonstrate that while mRNA levels of *Ceacam1* remained unchanged among groups (Figure 5a), those mice supplemented with vitamin D exhibited elevated expression of *Ifnb1*, *Isg15*, and *Isg20* following MHV-3 infection (Figure 5b–d). This finding suggests that an enhanced IFN antiviral response could be a potential mechanism for safeguarding mice from severe lung damage induced by MHV-3 infection.

### 3.5. Vitamin D Reduces Extrapulmonary Damage Caused by MHV-3

MHV-3 is known to be a hepatotropic virus and causes severe liver disease in mice [48]. We have previously demonstrated that MHV-3 delivered via the intranasal route also spread to distant organs in C57BL/6 mice, mainly triggering fatal hepatitis from 5 dpi onwards [24]. Here, we investigated the effect of vitamin D on the systemic manifestation of MHV-3 infection. Large amounts of infectious virus were recovered on the 5th dpi from liver homogenates of all MHV-3-infected mice fed a standard diet (Figure 6a). In animals fed a vitamin D-rich diet before infection, only 50% (n = 3/6) had detectable infectious virus in the liver, albeit at titers comparable to those of the STD group (*p* = 0.0634; Figure 6a). The gross morphology of the liver was distinguishable between groups. While the liver lobes of infected mice in the STD group exhibited a “cirrhotic/fibrotic” appearance at 5 dpi, those in the Vit. D group had an appearance similar to mock controls (Figure 6b). As expected, histologic evidence of severe liver disease was more pronounced in MHV-3-infected mice in the STD group, with apparent hepatitis, liver necrosis, and hydropic degeneration (Figure 6c,d). In the Vit. D group, animals developed milder but still significant alterations at 5 dpi, with small foci of necrosis and polymorphonuclear cells sparsely distributed in the hepatic tissue, besides mild hydropic degeneration (Figure 6c,d). These results indicate that vitamin D alleviates the deleterious effects of MHV-3 on the liver, which is in line with the improved survival rate seen in the Vit. D group (Figure 2d).

## 4. Discussion

The intricate interplay between vitamin D status and COVID-19 has been a topic of debate since the early days of the pandemic. Numerous observational studies have established links between hypovitaminosis D and an elevated risk of SARS-CoV-2 infection or worse outcomes (see meta-analysis by [19,49]). Conversely, clinical trials exploring vitamin D supplementation, either as a preventive or therapeutic measure for COVID-19, have yielded divergent results [17,18]. Variations in study designs, patient demographics, and vitamin D intervention protocols pose substantial challenges to drawing definitive conclusions in this context. Furthermore, there still remains a dearth of preclinical evidence to support or refute the hypothesis that vitamin D, whether at normal or augmented levels, may confer protection against SARS-CoV-2 infection or any other β-CoV. This focal study contributes to advancing knowledge regarding this matter, as it demonstrates that vitamin D in its bioactive form (calcitriol) significantly reduces the in vitro replication of SARS-CoV-2, MHV-3, and MHV-A59. This finding is consistent with a recent study reporting significant antiviral activity of calcitriol in SARS-CoV-2-infected primary human nasal epithelial cell lines [23]. Nonetheless, such protective activity awaits elucidation in vivo. To the best of our knowledge, the only two other preclinical studies on this topic have demonstrated that both vitamin D and calcitriol failed to show antiviral effects against SARS-CoV-2 respiratory infection in vivo, as interrogated in the commonly used K18-hACE2 transgenic mouse model and Syrian hamster [22,23].

In response to this gap, we performed a comprehensive examination of vitamin D’s effects on wild-type mice intranasally exposed to MHV-3. This is a murine coronavirus that naturally infects mice and serves as a prototype to study pulmonary and systemic alterations commonly observed in moderate/severe cases of COVID-19 [24,25,26]. Our results indicate that dietary vitamin D supplementation two weeks prior to MHV-3 challenge greatly decreases the acute damage observed in the lungs as well as in extrapulmonary sites, leading to improved survival rates. The dose of vitamin D chosen in this study (10,000 IU/kg diet) has proven to be safe for mice and reduces inflammation-associated damages in the lung [50,51,52], besides enhancing diaphragm strength [53] and inhibiting the growth of lung cancer xenografts [54]. However, our supplementation protocol differs from those adopted in the aforementioned studies reporting limited or no benefits of vitamin D and calcitriol in murine models of SARS-CoV-2 infection [22,23]. Furthermore, it is worth noting that our findings highlight a prophylactic benefit of vitamin D supplementation and cannot be used to predict a therapeutic effect of this compound against murine or human coronaviruses, as explored in the in vitro studies. In this case, assessing the therapeutic effects of vitamin D in vivo can be challenging, as the absorption and conversion to calcitriol times may not align with the limited treatment window in current murine experimental models (usually <6 dpi). While one approach to address this issue is the administration of calcitriol instead of vitamin D itself, as adopted in the Mok et al. 2023 [23] protocol, the calcitriol dose may be too low to produce any effects due to its hypercalcemic nature. Therefore, future studies in this regard could benefit from focusing on treatment with other VDR agonists [55], which can be administered at a potentially effective dose.

The immune response against coronaviruses critically depends on type I interferon (IFN-I), which engages both the innate and adaptive immune systems. Early MHV control relies on IFN-I-mediated defense mechanisms [56,57]. In the context of SARS-CoV-2, deficiencies in IFN-I responsiveness significantly enhance COVID-19 severity [58,59,60]. Once β-CoVs have adapted to evade the antiviral effects of IFNs, the impact of interferon activity is most significant during the initial stages of the immune response [61,62,63]. Our findings showed that vitamin D supplementation prior MHV-3 infection enhanced the expression of IFN-β and the interferon-stimulated genes *Isg15* and *Isg20* in the lungs upon infection. Elevated levels of IFN-β and ISGs have been shown to potentiate the host antiviral response [7,39,60] and likely explain the significant reduction of MHV-3 titers in the lungs and plasma. Another important means by which vitamin D enhances host’s defense is increasing the synthesis of antimicrobial peptides within the lungs, notably LL-37. Indeed, LL-37 administration robustly inhibits SARS-CoV-2 replication both in vitro and in vivo [21]. Nevertheless, it is imperative to note that this study did not focus in this direction, primarily due to the fact that vitamin D response elements (VDRE) within the promoter regions of genes responsible for encoding LL-37 and other important antimicrobial peptides are not conserved in rodents [2,4].

Murine coronavirus and SARS-CoV-2 trigger similar host inflammatory responses, leading to immune cell infiltration in the bronchioalveolar space and lung interstitial compartment, besides excessive amounts of pro-inflammatory cytokines [37,39,64]. Several cytokines have been shown to be implicated in the deleterious effects induced by β-CoVs on the respiratory system [64,65], with a major protagonism for TNF and IFN-γ [44]. It has been shown that the synergism between increased TNF and IFN-γ triggers a wave of massive cell death known as PANoptosis [44,47]. Moreover, the respiratory and systemic damage caused by MHV and SARS-CoV-2 is highly influenced by the activation of NLRP3 inflammasome [66,67,68]. In our experimental condition, vitamin D supplementation was shown to either reduce or prevent the MHV-3-induced increase in pulmonary concentration of TNF and IFN-γ, as well as IL-6 and the active inflammasome-dependent cytokine IL-1β. While our analysis did not encompass various stages of MHV-3 infection, as previously described [24], the findings are in concordance with the robust decrease in immune cell infiltration, cell death rate, and injury score observed in the lung tissue of infected mice in the Vit. D group at the peak of infection (3 dpi). This observation underscores a potential connection between cytokine modulation and the mitigated pathological effects on the lungs of MHV-3-infected mice treated with Vitamin D.

### Limitations

Despite the promising findings, this study has notable limitations. The use of the MHV-3 infection mouse model, while advantageous for studying fundamental aspects of β-CoV-induced disease in a biosafety level 2 environment [24,25,26,69], introduces potential disparities in pathogenesis compared to SARS-CoV-2, including differences in host cell entry receptors. Incorporating SARS-CoV-2 into future in vivo experiments is crucial for a more accurate assessment of vitamin D’s potential benefits in the context of COVID-19. Furthermore, the inclusion of control groups receiving a standard diet (1000 IU vitamin D_3_/kg diet), rather than reflecting the prevalent vitamin D deficiency observed in a substantial portion of the human population [70], limits the generalizability of our findings and comparisons with other relevant preclinical studies [22]. Considering the inflammatory bone loss observed in both MHV-3 infection in mice [24,25] and SARS-CoV-2 models [71], maintaining a group on a vitamin D-deficient diet could have intensified an already severe phenotype in a serious disease, challenging certain comparisons. Future research should address this limitation by including vitamin D-deficient mice to shed light on the potential interplay between vitamin D deficiency and coronavirus susceptibility.

## 5. Conclusions

This study contributes to expanding the realm of research delving into the intricate interplay between vitamin D and coronavirus infections. The observed significant reduction in in vitro replication of SARS-CoV-2, as well as murine coronaviruses MHV-3 and MHV-A59, underscores the potential antiviral properties of vitamin D against β-CoVs. Notably, our in vivo investigation highlighted a pronounced role for vitamin D supplementation in attenuating inflammatory lung damage in a mouse model of SARS-like disease induced by MHV-3. While these findings open promising avenues for further exploration and emphasize the importance of continued research in this area, it is imperative for future studies to broaden the scope of this study by encompassing in vivo infection with SARS-CoV-2. This imperative step will provide a more comprehensive understanding of vitamin D’s prospects in addressing COVID-19. As a result, we argue that our findings should not be construed in the context of the current clinical landscape of studies on vitamin D and COVID-19. Instead, they may serve as a springboard for future preclinical research aimed at exploring the intricate relationship between Vitamin D and β-CoV infections.

## Figures and Tables

**Figure 1 viruses-15-02434-f001:**
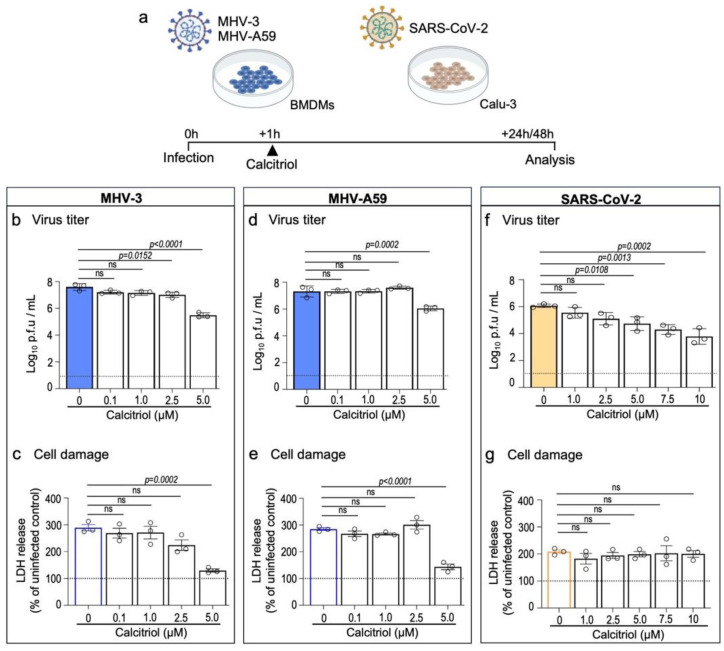
Calcitriol, the bioactive metabolite of vitamin D, inhibits in vitro replication of MHV-3, MHV-A59, and SARS-CoV-2. (**a**) Experimental design. BMDMs were infected with 0.1 MOI of MHV-3 or MHV-A59 and treated for 24 h with increasing concentrations of calcitriol. Infection with SARS-CoV-2 (0.1 MOI) was established in the human airway cell line Calu-3 48 h prior to treatment with calcitriol. (**b**,**d**,**f**) MHV-3, MHV-A59, and SARS-CoV-2 titers were determined using plaque assay on cell supernatants. Limit of detection threshold = 1 Log10 p.f.u/mL. (**c**,**e**,**g**). Cell damage analysis based on lactate dehydrogenase (LDH) release/activity in the supernatant samples specified in (**b**,**d**,**f**). Differences between the vehicle (0 μM) and calcitriol treatment groups were assessed via one-way ANOVA plus Dunnett’s test. ns: not statistically significant (*p* > 0.05). n = 3 independent experiments.

**Figure 2 viruses-15-02434-f002:**
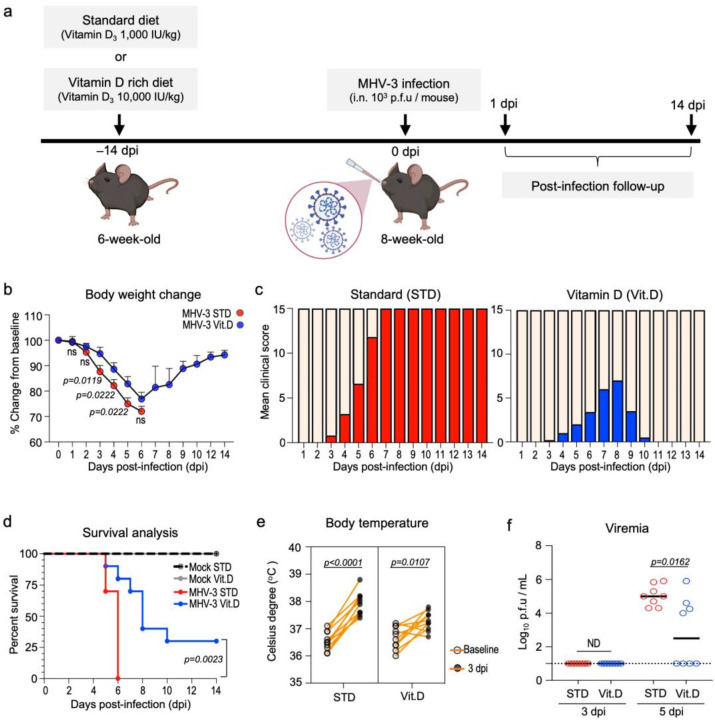
Dietary vitamin D supplementation has protective effects in wild-type mice upon lethal MHV-3 infection. (**a**) Experiment design: 6-month-old mice were either fed a standard diet or a vitamin D-rich diet for 2 weeks prior to intranasal instillation of MHV-3. Mice were monitored daily for signs of disease for up to 14 days post infection (dpi). (**b**) Body weight changes over the course of infection. Data are represented as mean ± S.E.M. Differences were assessed from 1 to 6 dpi via two-way ANOVA plus Fisher’s LSD test. n = 7–10. (**c**) Evolution of clinical symptoms between groups over the course of infection. Mean clinical scores encompass pooled differences in body weight as well as the presence and grade of hunched posture, ruffled fur, lethargy, and rapid shallow breathing (see materials and methods). n = 10 (**d**) Kaplan–Meier survival curve of mock- and MHV-3-infected mice according to diet. n = 10. (**e**) Pairwise t-test comparison of the febrile response triggered by MHV-3 infection in mice at 3 dpi. Body temperature was registered rectally. n = 8. (**f**) MHV-3 load determined via plaque assay in the plasma of infected mice following 3 and 5 dpi. Limit of detection threshold = 1 Log10 p.f.u/mL. Differences between groups were analyzed via the Mann–Whitney U test. n = 8. ND: not determined. STD: standard diet group. Vit. D: vitamin D-rich diet group.

**Figure 3 viruses-15-02434-f003:**
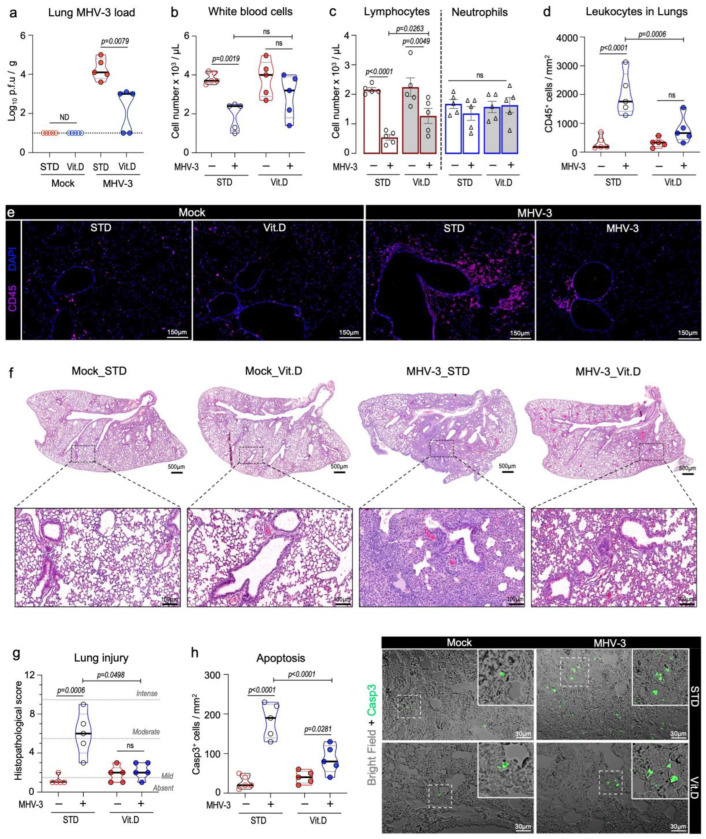
Vitamin D mitigates inflammation and acute lung damage caused by MHV-3 at 3 dpi. (**a**) Viral load measured via plaque assay in lung tissue extracts of MHV-3-infected mice. Limit of detection threshold = 1 Log_10_ p.f.u/g. (**b**,**c**) Changes in the number of total circulating leukocytes as well as lymphocytes and neutrophils among groups. Cell densities were determined in blood samples via an automatic hemocytometer. (**d**) Quantification of leukocyte density per lung area (mm^2^) among groups. Leukocytes were stained with CD45 and analyzed via confocal microscopy. (**e**) Representative images depicting the group aspects of CD45^+^ cell distribution throughout the lung tissue. (**f**) Representative images of mouse lung sections stained with hematoxylin and eosin, depicting overall histological differences among experimental groups. (**g**) Comparison of the lung injury score among groups. (**h**) Quantification of apoptotic cells in mouse lung sections, as detected via cleaved caspase 3 staining. Differences between mock and infected mice in relation to diet protocol were analyzed via ordinary one-way ANOVA plus Fisher’s LSD test. ns: non-statistically significant. ND: not determined. STD: standard diet group. Vit. D: vitamin D-rich diet group. n = 5.

**Figure 4 viruses-15-02434-f004:**
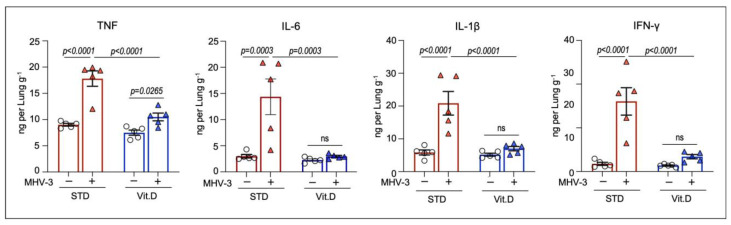
Vitamin D limits intrapulmonary overload of TNF, IL-6, IL-1β, and IFNγ following MHV-3 infection. Cytokine concentrations were determined in lung homogenates via enzyme-linked immunosorbent assay (ELISA) at 3 dpi. Differences between experimental groups were assessed via ordinary one-way ANOVA plus Fisher’s LSD test. ns: non-statistically significant. n = 5.

**Figure 5 viruses-15-02434-f005:**
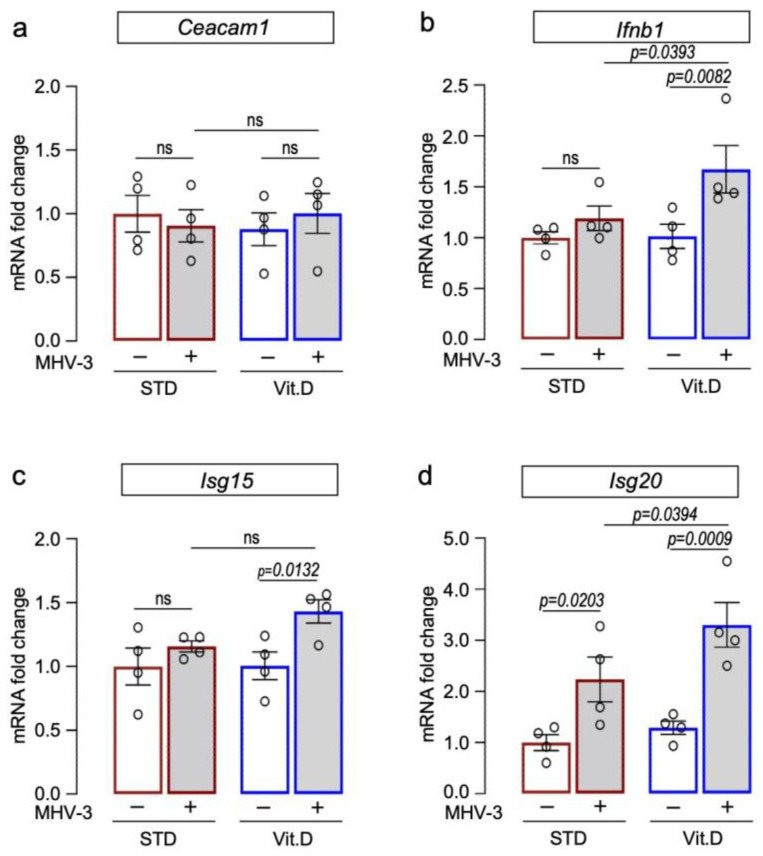
Vitamin D supplementation potentiates expression of interferon beta 1 (Ifn1b) and interferon-stimulated genes in lungs of MHV-3-infected mice at 3 dpi. Expression of target genes encoding for the MHV entry receptor Ceacam 1 (**a**), interferon beta 1 ((**b**); *Ifnb1*), and interferon-stimulated gene 15 ((**c**); *Isg15*) and 20 ((**d**); *Isg20*) were determined via qRT-PCR. mRNA levels were analyzed via 2^−ΔΔCT^ using *Gapdh* and *Actb* as reference genes. Ordinary one-way ANOVA plus Fisher’s LSD test were used to assess differences among groups. ns: non-statistically significant. n = 5.

**Figure 6 viruses-15-02434-f006:**
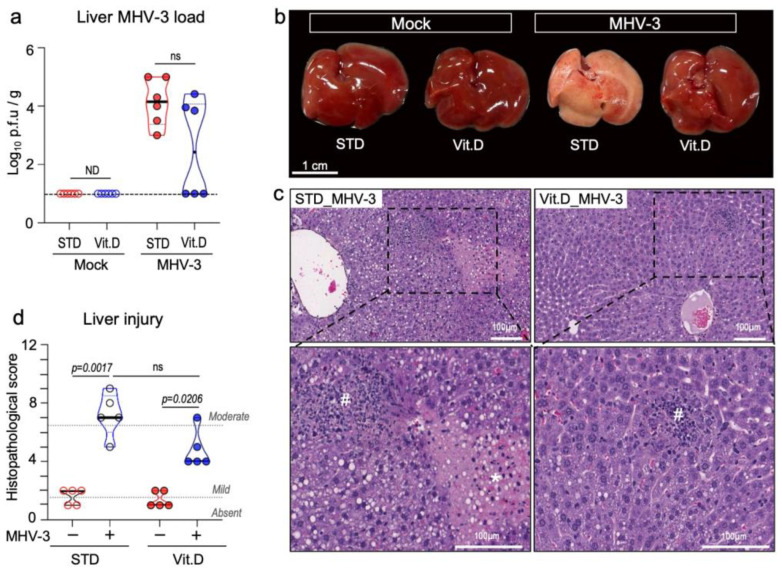
Vitamin D alleviates hepatic damage caused by MHV-3 respiratory infection at 5 dpi. (**a**) MHV-3 load determined in liver tissue homogenates via plaque assay. Limit of detection threshold = 1 Log_10_ p.f.u/g. Comparisons were made using the Mann-Whitney U test. (**b**) Representative images of the gross morphologic aspects of the liver across groups. (**c**) Representative images of the main changes observed in the liver parenchyma upon MHV-3 challenge. #: indicates inflammation foci. *: indicates necrotic areas. (**d**) Mean injury scores as determined in liver tissue sections stained with hematoxylin and eosin. Differences between groups were assessed via Kruskal–Wallis plus uncorrected Dunn’s test. ns: non-statistically significant. ND: not determined. STD: standard diet group. Vit. D: vitamin D-rich diet group. n = 5–6.

**Table 1 viruses-15-02434-t001:** List of primers used for RT-qPCR.

Gene	NCBI Reference	Forward Primer	Reverse Primer
*Gapdh*	NM_001289726.2	5′-AGGTCGGTGTGAACGGATTTG-3′	5′-TGTAGACCATGTAGTTGAGGTCA-3′
*Actb*	NM_007393.5	5′-ATGTTTGAGACCTTCAACA-3′	5′-CACGTCAGACTTCATGATGG-3′
*Ceacam1*	NM_001039185.1	5′-CTTGGAGCCTTTGCCTGGTA-3′	5′-ATCTCTCTGCCGCTGTATGC-3′
*Ifnb1*	NM_010510.2	5′-ACTCATGAAGTACAACAGCTACG-3′	5′-GGCATCAACTGACAGGTCTT-3′
*Isg15*	NM_015783.3	5′-CTGCAGCAATGGCCTGGGACCT-3′	5′-AGTTTGGTGGGCCAGGCGCT-3′
*Isg20*	NM_001113527.1	5′-GCCTGGAGGGCTGTTGGTTCTTG-3′	5′-CTGCCATGCTCCTTGGCGACC-3′

**Table 2 viruses-15-02434-t002:** Assessing the effects of 2 weeks vitamin D supplementation in mice.

Variable	Standard Diet (1000 IU Vitamin D_3_/kg Diet)	Vitamin D Rich Diet (10,000 IU Vitamin D_3_/kg Diet)	*p* Value
Serum 25(OH)D	24.75 ± 3.39 ng/mL	48.84 ± 4.80 ng/mL	0.0060
Serum calcium	5.724 ± 0.35 mg/dL	6.870 ± 0.57 mg/dL	0.1230
Mean food intake per mouse			
Week 1	2.312 ± 0.09 g/day	2.434 ± 0.09 g/day	0.3659
Week 2	2.545 ± 0.23 g/day	2.651 ± 0.02 g/day	0.6584
Body weight (% from baseline)			
Baseline	17.52 ± 0.55 g (100.0)	17.86 ± 0.59 g (100.0)	0.6783
Week 1	17.42 ± 0.50 g (99.43)	17.41 ± 0.43 g (97.48)	0.9907
Week 2	18.30 ± 0.63 g (104.4)	17.90 ± 0.46 g (100.2)	0.6265

Data are mean ± S.E.M. Comparisons were made via unpaired *t*-test. n = 5.

## Data Availability

Data are contained within the article and Appendix A.

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
