# Peer review of "Dietary Vitamin D Mitigates Coronavirus-Induced Lung Inflammation and Damage in Mice"

_viruses, 2023, doi:10.3390/v15122434_

Round 1

Reviewer 1 Report

Comments and Suggestions for Authors

1.       Why authors did this study while several studies are already been published about COVID-19 and Vitamin D?

2.       There are standard clinical trials published that suggest Vitamin D does not help reduce COVID-19 risk. Here is a reference https://jamanetwork.com/journals/jama/fullarticle/2797574. Please discuss and justify.

3.       Authors need to include a table about the clinical trials testing Vitamin D for COVID-19 and discuss both the pros and cons of obtained results, according to the current study.

4.       Please include a conclusions section separately.

5.       Also, authors need to include a separate section describing the limitations of the current study. 

Comments on the Quality of English Language

This is fine. 

Author Response

We would like to acknowledge the invaluable contribution of the three reviewers who dedicated their time and expertise to thoroughly assess and enhance the quality of our scientific article during the peer-review process. 

1) Why authors did this study while several studies are already been published about COVID-19 and Vitamin D?

We appreciate the opportunity to address this question and provide further clarification on the rationale of our study. Despite numerous studies on Vitamin D and COVID-19, the scientific community is still grappling with inconclusive evidence. The ambiguity arises from conflicting results in randomized clinical trials (RCTs) and observational studies, as explored in the introduction and discussion sections of our revised manuscript (Lines 57-76; 407-414). This ongoing debate has been prominently featured in editorials and opinion articles over the past three years in prestigious journals such as The Lancet (PMID: 33444566; PMID: 32758429) and Nature Reviews (PMID: 35869321), and has been thoroughly examined elsewhere, notably in the comprehensive review by Dr. Adrian Martineau (PMID: 36366796).

While important RCTs are underway to provide more clarity on the topic (NCT04536298, NCT04868903, NCT04482673, NCT04385940, and NCT03188796), the lack of preclinical evidence hinders a comprehensive understanding of Vitamin D's potential benefits in the context of coronavirus infection. In this study, we leverage our expertise in basic animal science to investigate whether elevated levels of Vitamin D could confer protection in a preclinical model of β-CoV respiratory infection. It is important to emphasize that our objective is not to resolve the inconsistencies observed in RCTs or observational studies. Instead, our focus is on contributing novel preclinical evidence to underscore the significance of continued research in this area. By doing so, we aim to stimulate further inquiry and exploration into the intricate relationship between Vitamin D and β-CoV infection, recognizing that answers may lie beyond the current clinical landscape.

2) There are standard clinical trials published that suggest Vitamin D does not help reduce COVID-19 risk. Here is a reference https://jamanetwork.com/journals/jama/fullarticle/2797574. Please discuss and justify.

As rightly highlighted by the reviewer, there exists a body of studies suggesting that Vitamin D levels or supplementation may not necessarily reduce the risk of COVID-19. In our original manuscript, we conscientiously cited some of these studies, particularly in the introduction (2nd and 3rd paragraph) and discussion (1st paragraph). Notably, the JAMA publication mentioned by the reviewer comments on the CORONAVIT study conducted by Dr. Adrian Martineau's group and published in the BMJ journal in 2022 (PMID: 36215226; Ref.18). This open-label phase 3 RTC in the UK found no significant association between Vitamin D supplementation over six months and protection against COVID-19 or other acute respiratory tract infections.

However, it is crucial to approach the results of the CORONAVIT study with caution due to several identified limitations, notably the allowance of vitamin D supplement consumption among participants in the control arm. The same research group has suggested that the beneficial effects of vitamin D against acute respiratory tract infections are optimal when vitamin D supplements are taken for up to one year (PMID: 33798465). On the opposite direction, a separate RTC conducted with COVID-19 hospitalized patients in Spain reported that only 2% of participants receiving oral 25(OH)D administration required intensive care, compared to 50% in the randomized control arm (PMID: 32871238). This example serves to illustrate the ongoing debate and uncertainty surrounding the association between Vitamin D and COVID-19.

We acknowledge that the divergent results observed across studies likely stem from variations in study designs, patient demographics, and Vitamin D intervention protocols, which have been thoroughly addressed in the discussion section of our revised manuscript.

3) Authors need to include a table about the clinical trials testing Vitamin D for COVID-19 and discuss both the pros and cons of obtained results, according to the current study.

We appreciate your valuable suggestion and acknowledge the significance of contextualizing our current study within the broader landscape of clinical research conducted thus far. While our primary objective does not involve presenting a comprehensive table summarizing individual clinical trials — recognizing that such an endeavor is more fitting for a review article — we have thoughtfully incorporated references to pertinent systematic reviews and meta-analyses (line 72-73; Refs.19 and 20). These reviews provide current and valuable insights, offering a nuanced understanding of both the positive and negative effects of Vitamin D supplementation in the specific context of COVID-19.

In addition, given the notable limitations of our experimental models, we have argued that our findings should not be construed in the context of the current clinical landscape of studies on vitamin D and COVID-19. Instead, they may serve as a springboard for future preclinical research aimed at exploring the intricate relationship between Vitamin D and β-CoV infections. See lines 510-515.

4) Please include a conclusions section separately.

We have included a dedicated conclusions section to encapsulate the key findings and implications of our research (Lines 500-515).

5) Also, authors need to include a separate section describing the limitations of the current study.

We have thoroughly discussed the primary limitations of the present study within a dedicated subsection of the discussion (Lines 483-498).

Reviewer 2 Report

Comments and Suggestions for Authors

This manuscript explores the prophylactic capability of VitD during CoV infection. Calcitriol, the active form of Vit D was shown to inhibit viral adsorption and host-cell entry of SAR-CoV-2. However, it did not confer any protection in vivo. This study is the follow up of that work where the authors have tested the ability of Vit D to confer protection using MHV-3 as a CoV model. In this paper, the authors show that calcitriol reduces SARS-CoV-2 and MHV-2 replication in vitro and that Vit D supplementation in animal diet reduces lung inflammation, tissue damage and significantly reduced the lethality of mice. I definitely recommend this paper for publication as it is well written and is an important study adding significant insight to Vit D possibly being an important factor during CoV infection.

Minor concerns

1)      Line 261-262, since fig 2D also talks about mortality, that can also be added in the sentence.

2)      Line 316: ig3 legend lacks the days post infection info. Even though this information in added in the text, it would be more consistent to add it to the legend too as other figure legends have this info.

3)      All the data shown are from experiment where mice are pretreated with VitD for 14 days before infection. If Vit D is to be considered a potent prophylactic drug, the possibility of addition of Vit D atleast a few days post infection or what effect it might have must be explore, at least discussed.     

Author Response

We would like to acknowledge the invaluable contribution of the three reviewers who dedicated their time and expertise to thoroughly assess and enhance the quality of our scientific article during the peer-review process.

1) Line 261-262, since fig 2D also talks about mortality, that can also be added in the sentence.

We have modified the text accordingly (lines 261-262).

2) Line 316: ig3 legend lacks the days post infection info. Even though this information in added in the text, it would be more consistent to add it to the legend too as other figure legends have this info.

We appreciate your suggestion. All analyses presented in figure 3 were carried out at 3pi and this information is now part of the figure legend (Line 315).

3) All the data shown are from experiment where mice are pretreated with VitD for 14 days before infection. If Vit D is to be considered a potent prophylactic drug, the possibility of addition of Vit D at least a few days post infection or what effect it might have must be explore, at least discussed.

We appreciate your insightful comment. While our current study primarily emphasizes the prophylactic effects of Vitamin D, we acknowledge the significance of exploring post-infection interventions. It is important to note that our dietary intervention extends to the first week post-infection, as detailed in the materials and methods of the original manuscript (Line 151). This timeframe corresponds to the critical period of MHV infection. Although our study design does not specifically align with therapeutic goals, we recognize this as a valuable area for further investigation. We have underscored this aspect in our ongoing discussions (lines 438-448).

Reviewer 3 Report

Comments and Suggestions for Authors

Opinions on the effect of vitamin D on the pathogenesis of COVID-19 are currently divided. So the results, which may of course be in vitro or in experimental animals, are still valuable. The results of the study are of course in vitro or in experimental animals, but they are valuable nonetheless, because they may lead to a more reliable treatment or even to a better treatment. Therefore, it is very regrettable that all the experiments on mice were with MHV-3 virus. This would ultimately lead to results that are irrelevant to the COVID-19 pandemic.

 A systematic review and meta-analysis of the effect of vitamin D on the pathogenesis of COVID-19 has been conducted. See these reviews in the introduction, which summarise the results of the questionable and the positive effects, respectively.

https://www.frontiersin.org/articles/10.3389/fnut.2023.1131103/full

https://pubmed.ncbi.nlm.nih.gov/35166850/

The author seems to be quite concerned about multiplicity in tests, but I think that multiplicity is not a major concern in these experimental systems. In the first place, P(H0) is very low, and multiplicity only becomes a problem when it is close to 1. Instead, the P-value obtained should be indicated instead of a threshold value, such as P<0.05. This is because this is another piece of information obtained. Table 1 is highly commendable in this respect.

Is 10000 IU/Kg the amount contained in the feed? I was confused at first because I thought this was the amount per mouse weight, which is a very high amount. I would be grateful if the author could work out how to write this.

The mice on the standard diet don't seem to be particularly deficient either, quite a lot of people are deficient in this, which probably makes them more susceptible to severe COVID. What would happen if this value were set lower? Would the effect of slowing down the growth of the virus identified in vitro also be seen in vivo?

I would then like the author to try SARS-CoV-2 using susceptible mice such as K18-hACE2. If this cannot be done, then the entire introduction, which is based on COVID-19, should be rewritten, and the value of this paper will be much less.

Author Response

We would like to acknowledge the invaluable contribution of the three reviewers who dedicated their time and expertise to thoroughly assess and enhance the quality of our scientific article during the peer-review process.

Below, you will find our responses to your comments. 

Commentary 1 (grouped): Opinions on the effect of vitamin D on the pathogenesis of COVID-19 are currently divided. So the results, which may of course be in vitro or in experimental animals, are still valuable. The results of the study are of course in vitro or in experimental animals, but they are valuable nonetheless, because they may lead to a more reliable treatment or even to a better treatment. Therefore, it is very regrettable that all the experiments on mice were with MHV-3 virus. This would ultimately lead to results that are irrelevant to the COVID-19 pandemic.

 A systematic review and meta-analysis of the effect of vitamin D on the pathogenesis of COVID-19 has been conducted. See these reviews in the introduction, which summarise the results of the questionable and the positive effects, respectively.

 https://www.frontiersin.org/articles/10.3389/fnut.2023.1131103/full

https://pubmed.ncbi.nlm.nih.gov/35166850/

 I would then like the author to try SARS-CoV-2 using susceptible mice such as K18-hACE2. If this cannot be done, then the entire introduction, which is based on COVID-19, should be rewritten, and the value of this paper will be much less.

We value your insightful comments and remain committed to enhancing the clarity and understanding of our research. Additional references of systematic reviews and meta-analysis have been added in the revised manuscript (Refs. 19 and 20) and the main text has been modified to highlight the ongoing debate on vitamin D and covid-19 (Lines 57-86) as well as the limitations of our study (Lines 483-498).

We appreciate and acknowledge your concerns regarding the use of the MHV-3 virus in our mouse experiments and the potential limitations in directly applying these results to the COVID-19 pandemic. Your suggestion to employ the hK18-SARS-CoV-2 mouse model is well-received, and we are committed to exploring this model in future research. To enhance the generalizability of our findings, we outlined the significant limitations imposed by the in vivo experimental model in a dedicated subsection of the discussion (Lines 483-498).

While we recognize the suitability of murine coronaviruses as SARS-like models, we acknowledge the need for cautious interpretation of our in vivo findings. Nevertheless, we believe these findings may offer valuable insights into the intricate relationship between Vitamin D and β-CoV infection.

Commentary 2: The author seems to be quite concerned about multiplicity in tests, but I think that multiplicity is not a major concern in these experimental systems. In the first place, P(H0) is very low, and multiplicity only becomes a problem when it is close to 1. Instead, the P-value obtained should be indicated instead of a threshold value, such as P<0.05. This is because this is another piece of information obtained. Table 1 is highly commendable in this respect.

Thank you for your comments on multiplicity concerns in our experimental systems. The use of "ns" in figures for P > 0.05 aims to enhance visual clarity, particularly with multiple comparisons, without compromising the comprehensive reporting in our supplementary files. In these files, we include all raw data and full P-values for every comparison, aligning with our commitment to transparency and ensuring the accessibility of every detail in our findings to the scientific community.

Commentary 3: Is 10000 IU/Kg the amount contained in the feed? I was confused at first because I thought this was the amount per mouse weight, which is a very high amount. I would be grateful if the author could work out how to write this.

Thank you for bringing this to our attention. Yes, the concentration of 10,000 IU/Kg refers to the amount of vitamin D3 per kg of chow. We have clarified this in the text (Lines 249,250,433,491) and table 2.

Commentary 4: The mice on the standard diet don't seem to be particularly deficient either, quite a lot of people are deficient in this, which probably makes them more susceptible to severe COVID. What would happen if this value were set lower? Would the effect of slowing down the growth of the virus identified in vitro also be seen in vivo?

We appreciate your insightful question. We acknowledge that our experimental design did not specifically aim to assess vitamin D deficiency but rather its supplementation. This decision was primarily based on the severe infection model used, characterized by normal vitamin D levels in the standard diet.

As demonstrated in our initial publication (PMID: 34495692) and subsequent works, severe MHV-3 infection leads to inflammatory bone loss (PMID: 37142087), a phenomenon also observed in SARS-CoV-2 models (PMID: 35534483). Therefore, we believe that maintaining a group of animals on a vitamin D-deficient diet could have resulted in a more severe phenotype in an already grave disease. This might have included even more abrupt fatalities, making certain comparisons challenging. We remain committed to enhancing the clarity and understanding of our research, and additional text addressing this matter has been incorporated into the revised version of the manuscript (Lines 490-498).

Round 2

Reviewer 1 Report

Comments and Suggestions for Authors

The authors successfully responded to the reviewer's comments and updated the manuscript as well. 

Comments on the Quality of English Language

This is fine. 

Reviewer 3 Report

Comments and Suggestions for Authors

It is unfortunate that it was not possible to find out how COVID-19 affects the mice. This is because the question now is how VitD affects the disease.

However, the fact that it does have a positive course in lung disease remains important. It is not in the reader's interest that I refuse to accept this. Introduction and 4.1. limitations are good revisions.

But again, I would encourage the authors to experiment with SARS-CoV-2. Alternatively, hamsters or minks could be used instead of mice. This team should be able to do it.